# Indicators to distinguish symptom accentuators from symptom producers in individuals with a diagnosed adjustment disorder: A pilot study on inconsistency subtypes using SIMS and MMPI-2-RF

Cristina Mazza[1☉], Graziella Orrù[2‡], Franco Burla[1‡], Merylin Monaro[3‡], Stefano Ferracuti[1‡], Marco Colasanti[1☉], Paolo Roma[1☉]*

1 Department of Human Neuroscience, Faculty of Medicine and Dentistry, Sapienza University of Rome, Rome, Italy, 2 Department of Surgical, Medical, Molecular & Critical Area Pathology, University of Pisa, Pisa, Italy, 3 Department of General Psychology, University of Padova, Padova, Italy

☉ These authors contributed equally to this work.
‡ These authors also contributed equally to this work.
* paolo.roma@uniroma1.it

**Data Availability Statement:** A repository for the data has been created in Zenodo. It can be

## Abstract

In the context of legal damage evaluations, evaluees may exaggerate or simulate symptoms in an attempt to obtain greater economic compensation. To date, practitioners and researchers have focused on detecting malingering behavior as an exclusively unitary construct. However, we argue that there are two types of inconsistent behavior that speak to possible malingering—accentuating (i.e., exaggerating symptoms that are actually experienced) and simulating (i.e., fabricating symptoms entirely)—each with its own unique attributes; thus, it is necessary to distinguish between them. The aim of the present study was to identify objective indicators to differentiate symptom accentuators from symptom producers and consistent participants. We analyzed the Structured Inventory of Malingered Symptomatology scales and the Minnesota Multiphasic Personality Inventory-2 Restructured Form validity scales of 132 individuals with a diagnosed adjustment disorder with mixed anxiety and depressed mood who had undergone assessment for psychiatric/psychological damage. The results indicated that the SIMS Total Score, Neurologic Impairment and Low Intelligence scales and the MMPI-2-RF Infrequent Responses (F-r) and Response Bias (RBS) scales successfully discriminated among symptom accentuators, symptom producers, and consistent participants. Machine learning analysis was used to identify the most efficient parameter for classifying these three groups, recognizing the SIMS Total Score as the best indicator.

accessed via this link: https://doi.org/10.5281/zenodo.3548270

**Funding:** The authors received no specific funding for this work.

**Competing interests:** The authors have declared that no competing interests exist.

# Introduction

*Psychic damage* (or psychological/ psychiatric damage) can be defined as an alteration of psychic integrity (i.e., a qualitative and quantitative change in psychic elements, including primary mental abilities, affectivity, defense mechanisms, and mood) [1]. Although it is considered biological damage, it is not limited to medically assessable pathology; rather, it involves both objective and subjective elements, linked to an individual's unique personal history [2]. In the context of legal damage evaluations, individuals obtain economic compensation based on an estimation of damage: the higher the estimate, the higher the indemnity received. A psychopathological condition that is often presented in this context is *adjustment disorder*. The 5th edition of the *Diagnostic and Statistical Manual of Mental Disorders* (DSM-5) lists the following diagnostic criteria for this disorder: presentation of emotional or behavioral symptoms within 3 months of a specific stressor; experience of more stress than would normally be expected in response to the stressful life event, and/or stress that causes significant problems in one's relationships, either at work or at school; and symptoms that are not the result of another mental health disorder or associated with healthy grieving [3]. In the medico-legal context, disorders associated with depression and anxiety, such as chronic adjustment disorder with mixed anxiety and depressed mood, are the most frequently simulated [4], at a rate of over 50% [5].

Forensic practitioners are trained to evaluate whether participants might simulate or accentuate distress or a psychic disorder in order to unjustly obtain greater compensation [6]. The DSM-5 defines such behavior as *malingering* and describes it as the "intentional production of false or grossly exaggerated physical or psychological symptoms, motivated by external incentives" (p. 726) [3]. Overall, malingering is a serious social problem that increases costs for society [7,8]. The precise incidence of the phenomenon is largely unknown; Young [9,10] pointed out that many studies relied on overinclusive criteria leading to an increase in prevalence, but in forensic and disability samples its prevalence has been reported to be between 15±15%.

Lipman [11] indicated four types of malingering: invention, or completely fabricating symptoms; perseveration, or fraudulently presenting formerly present symptoms that have since ceased; exaggeration, or making existing symptoms appear worse than they really are; and transference or attributing genuine symptoms to an unrelated cause or accident. Similarly, Resnick [12] described three potential subtypes of malingering: pure malingering, which involves completely fabricating symptoms; partial malingering, which involves exaggerating existing symptoms; and false imputation, which involves attributing symptoms to a cause that has little or no relationship to their development. Despite these various classifications of malingering, however, the behavior has mostly been considered a unitary construct, both theoretically and empirically, assimilating the aspects of invention and exaggeration. The literature has focused on the differences between simulators and honest respondents, while research focusing on exaggerators and their unique attributes has not been conducted, even though exaggerating behavior is thought to be much more frequent than that of invention and false imputation [13].

Researchers have developed instruments specifically designed to detect malingering and included, in personality inventories, scales with the same purpose. For example, the Structured Interview of Reported Symptoms (SIRS) [14,15] and its second edition (SIRS-2) [16], the latter being a 172-item test designed for the assessment of feigning, including a scale to assess defensiveness. SIRS has been considered the "gold standard" in assessing psychiatric malingering [17] and received extensive validations [18]. A study [19] on SIRS-2 indicated that, when compared to SIRS, it reaches a higher specificity (94.3% vs. 92.0%) but a lower sensitivity (36.8% vs. 47.4% among forensic patients). Another example is the Test of Memory Malingering

(TOMM) [20], a 50-item visual recognition test designed to help distinguish between malingerers and true memory impairments. Its usefulness in discriminating between bona-fide individuals and malingerers has been evidenced by a series of study, conducted with different types of participants (college students, patients with traumatic brain injury and hospital outpatients), different research designs (simulation and known-group) and different procedures for stimuli presentation (paper-and-pencil and computer) [21]. A further example is the Inventory of Problems-29 (IOP-29) [22], a 29-item easy-to-use measure of non-credible mental and cognitive symptoms. Studies using this instrument yielded encouraging results in the detection of malingering [23,24] and indicated that it can be used in a multimethod symptom validity assessment along with TOMM [25]. Another example is the Structured Inventory of Malingered Symptomatology (SIMS) [26] is a 75-item multi-axial self-report questionnaire validated with clinical-forensic, psychiatric, and non-clinical populations. In the literature, many authors have used the SIMS to discriminate between honest respondents and simulators, confirming its usefulness for this task [27–33]. Regarding SIMS, scales' sensitivity partly depends on the feigned condition. A study by Edens, Otto and Dwyer [27] evaluating SIMS scales' sensitivity for different conditions (e.g., depression, psychosis, cognitive impairment) indicated that, while most scales (except Psychosis scale) were sensitive to malingering regardless of the specific symptomatology, Total Score was overall the most sensitive indicator, correctly identifying 96.4% of all malingered protocols. A last example is the Minnesota Multiphasic Personality Inventory-2 Restructured Form (MMPI-2-RF) [34], which is a 338-item personality questionnaire that includes subscales designed to detect overreporting and response bias (Infrequent Responses, F-r; Infrequent Psychopathology Responses, Fp-r; Symptom Validity, FBS-r; Infrequent Somatic Responses, Fs; Response Bias, RBS). A recent meta-analysis [35] indicated that the most sensitive scale for feigned mental disorders is RBS (.93, cut-off ≥80), followed by FBS-r (.84, cut-off ≥80) and F-r (.71, cut-off ≥100); for feigned cognitive impairment the most sensitive scale is FBS-r (.88, cut-off ≥80), followed by RBS (.84, cut-off ≥80) and Fs (.66, cut-off≥80); lastly, for feigned medical complaints, the most sensitive scale is RBS (.94, cut-off≥80), followed by FBS-r (.69, cutoff≥80) and F-r (.59, cut-off≥100).

In an attempt to develop a strategy to distinguish between symptom accentuators and producers, we employed a promising tool in the field of lie-detection: machine learning (ML). ML can be defined as "the study and construction of algorithms that can learn information from a set of data (called a training set) and make predictions for a new set of data (called a test set). In other words, it consists of training one or more algorithms to predict outcomes without being explicitly programmed and only uses the information learned from the training set." In the literature, ML models have been recently used to discriminate between honest respondents and fakers in a variety of settings [36–38], with extremely promising accuracy, indicating that ML can, in fact, outperform traditional statistical methods.

The main purpose of this study was to identify helpful criteria to distinguish accentuators from symptom producers and consistent individuals. We hypothesized that symptom accentuators would have on SIMS and MMPI-2-RF selected validity scales higher scores compared to consistent participants, but lower scores than symptom producers.

## Materials and method

### Participants and procedure

Participants were 150 Italian individuals who had to undergo a mental health examination based on a judge's order, between January and December 2018, in the context of a lawsuit involving psychological injury. All were referred to the Laboratory of Clinical Psychology at Human Neuroscience Department, Faculty of Medicine and Dentistry, Sapienza University of

Rome, which is an academic reference center in psychiatric evaluation and psychological assessment. Specifically, the inclusion criteria were: (a) having been born and raised in Italy, and (b) having a clinical diagnosis of chronic adjustment disorder with mixed anxiety and depressed mood. On the other hand, the exclusion criteria were: (a) a psychiatric history prior to the accident, and (b) comorbidity with another psychiatric disorder. Forty percent of these diagnoses followed a road accident, 30% were a consequence of work-related accidents, 20% followed equally workplace harassment and stalking episodes, and 10% originated from domestic violence.

The distribution of participants ($N = 150$) to groups and the evaluation of their psycho-diagnostic profiles were conducted in three phases.

In the first phase, participants underwent a psychiatric interview and a psychological-clinical interview, blind. At the end of these interviews, the psychiatrist and clinical psychologist, also blind, completed an information sheet establishing the following: First (a), congruence between the documentation submitted considered suitable (e.g., psychopharmacological prescriptions, psychotherapeutic treatment certifications, illness certificates for work) and the diagnosis provided by a mental health professional with the required training to diagnose chronic adjustment disorder with mixed anxiety and depressed mood. Second (b), congruence between the manifestation of clinical and emotional symptoms during the interview (e.g. lowering of mood, crying or hopelessness, nervousness, agitation) and the diagnosis of chronic adjustment disorder with mixed anxiety and depressed mood. And third (c), congruence between the referenced symptomatology and the referenced impairment in day-to-day functioning in the participants' social, working, and other important areas (e.g., absenteeism from work, changes and difficulties in interpersonal relationships, complications regarding illness and treatments such as extension of hospital stay and decreased compliance with the recommended treatment regimen). The determination for each of these criteria was "congruent" or "incongruent". The information sheets were then delivered directly to the research coordinator. Until the end of this evaluation step, the two mental health professionals did not have any knowledge regarding the assessment made by the other colleague on the same participant (i.e., blind procedure). Whenever there was disagreement on one or more conclusions, the experts were required to justify their choice and reach an agreement. At first, experts disagreed on the evaluation of the second criterion (b) 11 times, while 23 cases involved mental professionals reaching different conclusions on the third (c) parameter. Participants for whom agreement was impossible ($N = 12$ out of 34 divergences) were excluded from the study. Further ($N = 6$) participants were excluded because they did not consent to the research. In the second phase, examinees were assigned to one of three groups on the basis of the experts' conclusions: (a) Consistent Participants (CP), which included individuals judged congruent on *all* three criteria; (b) Symptom Accentuators (SA), which included examinees judged congruent on criterion 1 but incongruent on *either* criterion 2 or 3; and (c) Symptom Producers (SP), which included members judged incongruent on *at least* two criteria. In the third phase, participants completed a test battery with the help of specialized technical staff, blind. Test scoring was performed via computer software.

The final sample was comprised of 132 participants (Table 1). The three groups differed in age [$F (2, 129) = 8.373$, $p < .001$] and educational level [$F (2, 129) = 4.240$, $p = .016$], but not gender composition [$F (2, 129) = 1.775$, $p = .191$].

The study was carried out with written informed consent by all participants, in accordance with the Declaration of Helsinki. It was approved by the local ethics committee (Board of the Department of Human Neuroscience, Faculty of Medicine and Dentistry, Sapienza University of Rome).

**Table 1. Demographic composition of the three research groups.**

| | | Consistent Participants (n = 49) | Symptom Accentuators (n = 44) | Symptom Producers (n = 39) |
|---|---|---|---|---|
| Gender | **M** (n) | 42 | 31 | 29 |
| | **F**(n) | 7 | 13 | 10 |
| Age M (SD) | | 48.82 (6.84) | 44.55 (13.30) | 39.59 (10.77) |
| Education (years) M (SD) | | 13.08 (3.05) | 11.50 (2.28) | 12.00 (2.61) |

## Materials

**Structured Inventory of Malingered Symptomatology (SIMS)** [26,39]. The SIMS is comprised of 75 items that describe implausible, rare, atypical, or extreme symptoms that bona fide patients tend not to present. The response options are on a dichotomous scale ("True" vs. "False"), and the measure aims at detecting feigned psychopathology [40]. The item responses are grouped into five main scales, addressing the validity of symptoms related to Psychosis (P; evaluates the degree to which participants report unusual and bizarre psychotic symptoms that are not typically encountered in psychiatric populations), Low Intelligence (LI; assesses the degree to which participants simulate or exaggerate intellectual deficits through low performance on simple items), Neurological Impairment (NI; evaluates illogical or atypical neurological symptoms), Affective Disorders (AF; evaluates the degree to which participants present atypical symptoms of depression and anxiety), and Amnestic Disorders (AM; evaluates the degree to which participants report memory deficits that are inconsistent with patterns of impairment seen in brain dysfunction or injury). The total number of implausible symptoms endorsed by the subject represents the Total Score (TS), which is the main symptom validity scale of the SIMS. Indeed, the five SIMS subscales were not designed to detect the overreporting of mental health problems, but to determine which types of psychopathology respondents tend to overreport when the SIMS Total Score is above the cut-off value [39]. Different cut-off values have been used in the literature (i.e., $\geq 14$ [25]; $\geq 17$ [39]; $\geq 19$ [32], and even $\geq 24$ [41]). A recent meta-analytic study encompassing 4,180 protocols supported the claim that the specificity of the SIMS may be unsatisfactory when the traditional cut-offs (i.e., $\geq 15$ and $\geq 17$) are adopted [40]. The Italian version of the SIMS was translated by La Marca, Rigoni, Sartori, and Lo Priore [42].

**Minnesota Multiphasic Personality Inventory-2 Restructured Form (MMPI-2-RF)** [34]. The MMPI-2-RF is a 51-scale measure of personality and psychopathology with 338 items, selected from the 567 items on the complete MMPI-2. The response options are on a dichotomous scale ("True" vs. "False"). The MMPI-2-RF is comprised of: nine validity scales, most of which are revised versions of the MMPI-2 validity scales; nine Restructured Clinical (RC) scales, developed by Tellegen et al. and released in 2003 [43]; three Higher Order (HO) scales, derived from factor analyses to identify the basic domains of affect, thought, and behavior; 23 Specific Problem (SP) scales, intended to highlight important characteristics associated with particular RC scales; and revised versions of the Personality Psychopathology Five (PSY-5) scales, which link the MMPI-2-RF to a five-factor model of personality pathology [34]. The present study considered the following MMPI-2-RF validity scales: F-r, Fp-r, Fs, FBS-r, RBS, and K-r. The Italian version was translated by Sirigatti and Faravelli [44].

## Statistical analysis and machine learning models

A first multivariate analysis of variance with covariates (MANCOVA) was run using the three research groups (Consistent Participants, Symptom Accentuators, Symptom Producers) as the

independent variable and the SIMS Total Score and subscale scores as dependent measures. A second MANCOVA was run using the three research groups as the independent variable and MMPI-2-RF validity scale T-scores as dependent measures. Both analyses controlled for age and educational levels. The Bonferroni correction was applied to adjust confidence intervals; SPSS-25 software (SPSS Inc., Chicago, IL) automatically corrected the p-value for the number of comparisons. Scheffe's [45] method was used to assess post hoc pair differences ($p < 0.05$). The effect sizes of the score differences between groups was also measured; values of .02, .13, and .26 were considered indicative of small, medium, and large effects, respectively [46]. The SPSS-25 statistical package was used for all analyses. ML analyses were run using WEKA 3.8 [47].

## Results

### SIMS

A 3 x 6 MANCOVA (groups x SIMS scales) showed a significant effect of group on the selected SIMS scales, $V$ = .550, $F$ (12, 246) = 7.777, $p < .001$, par$\eta^2$ = .275. In more detail, the results for SIMS showed that the three research groups (Consistent Participants, Symptom Accentuators, Symptom Producers) obtained significantly different scores on the NI, LI, and TS scales. Furthermore, on the AF scale, there was a significant difference in scores between Consistent Participants and the other two groups. Lastly, there was a significant difference between Symptom Producers and the other participants on the P and AM scales. Table 2 shows the descriptive values of SIMS scores for each group.

### MMPI-2-RF

A 3 x 6 MANCOVA (groups x MMPI-2-RF selected scales) showed a significant effect of group on the MMPI-2-RF selected validity scales, $V$ = .377, $F$ (12, 246) = 4.758, $p < .001$, par$\eta^2$ = .188. In more detail, results for the MMPI-2-RF showed that the three groups (Consistent Participants, Symptom Accentuators, Symptom Producers) obtained significantly different scores on the F-r and RBS scales. Furthermore, on the Fp-r and Fs scales, a significant difference was found between the Symptom Producers and the other two groups. On the FBS-r scale, scores of the consistent group significantly differed from those of the accentuating and producing groups.

**Table 2. Comparison between consistent participants, accentuators, and symptom producers on SIMS mean scores.**

| SIMS | Consistent Participants $n$ = 49 Scores *M(SD)* | Symptom Accentuators $n$ = 44 Scores *M(SD)* | Symptom Producers $n$ = 39 Scores *M(SD)* | $F$ | $p$ | par$\eta^2$ |
|---|---|---|---|---|---|---|
| **Neurologic Impairment (NI)** | 1.31 (.94) A | 2.61 (2.26) B | 3.92 (2.21) C | 16.83 | < .001 | .210 |
| **Affective Disorder (AF)** | 5.73 (2.01) A | 8.14 (3.25) B | 8.46 (3.32) B | 11.24 | < .001 | .150 |
| **Psychosis (P)** | .90 (.90) A | 1.55 (1.39) A | 2.67 (2.18) B | 11.20 | < .001 | .150 |
| **Low Intelligence (LI)** | 1.06 (1.20) A | 2.11 (1.73) B | 4.31 (2.23) C | 32.92 | < .001 | .341 |
| **Amnestic Disorder (AM)** | 1.35 (1.17) A | 2.16 (1.84) A | 3.82 (2.78) B | 14.19 | < .001 | .183 |
| **Total Score (TS)** | 10.35 (4.05) A | 16.50 (5.77) B | 23.15 (6.29) C | 50.44 | < .001 | .443 |

*Note.* For each line, different letters indicate a significant difference between columns.

On the K-r scale, there was a significant difference between the consistent group and the producing group, but not between the accentuating group and either of the other groups. Table 3 shows the descriptive values of the MMPI-2-RF scales for each group.

## Feature selection and machine learning models

The recent focus on the lack of replicability in behavioral experiments has suggested that the discipline is facing a "replicability crisis." One potential source of this problem is the frequent use of inferential statistics with misunderstood $p$ values and underpowered experiments [48]. Recent methodological discussions relate to procedures that guarantee replicable results [49]. In summarizing their assessment of replicability, Szucs and Ioannidis [50] concluded that: "Assuming a realistic range of prior probabilities for null hypotheses, false report probability is likely to exceed 50% for the whole literature. In light of our findings, the recently reported low replication success in psychology is realistic, and worse performance may be expected for cognitive neuroscience" (p.1). The replication of experimental results may be distinguished according to exact versus broad replication [51]. Exact replication refers to replication that follows the exact same procedure of the original experiment, incorporating cross-validation. Cross-validation is generally a very good procedure for measuring the replicability of a given result. While it does not prevent model overfit, it still estimates true performance.

To avoid overfitting, cross-validation is regarded a compulsory step in ML analysis; nonetheless, its use is very limited in the analysis of psychological experiments. There are a number of cross-validation procedures, but one that consistently guarantees a good result is the so-called *k-fold* method (or *k*-fold cross-validation). The *k*-fold cross-validation is a technique used to evaluate predictive models by repeatedly partitioning the original sample into a training set to train the model, and a validation set to evaluate it. Specifically, in this paper, we adopted a 10-fold cross-validation procedure, in which the original sample was randomly partitioned into 10 equal-size subsamples, the folds. Of the 10 subsamples, a single subsample is retained as validation data for testing the model, and the remaining 10–1 = 9 subsamples are used as training data. Such process is repeated 10 times, with each of the 10 folds are then used exactly once as validation data. The results from the 10 folds were then averaged to produce a single estimation of prediction accuracy. Most psychometric investigations do not address the

**Table 3. Comparison between consistent participants, accentuators, and symptom producers on MMPI-2-RF selected validity scale mean scores.**

| MMPI-2-RF | Consistent Participants $n = 49$ Scores $M$ ($SD$) | Symptom Accentuators $n = 44$ Scores $M$ ($SD$) | Symptom Producers $n = 39$ Scores $M$ ($SD$) | $F$ | $p$ | $par\eta^2$ |
|---|---|---|---|---|---|---|
| **F-r** | 63.16 (8.18) A | 70.73 (12.64) B | 84.41 (13.98) C | 29.10 | < .001 | .314 |
| **Fp-r** | 58.84 (8.94) A | 62.86 (9.99) A | 72.64 (16.21) B | 11.56 | < .001 | .154 |
| **Fs** | 61.35 (12.06) A | 68.84 (17.57) A | 82.51 (19.26) B | 15.38 | < .001 | .195 |
| **FBS-r** | 56.33 (13.09) A | 65.41 (16.27) B | 72.74 (14.27) B | 11.71 | < .001 | .156 |
| **RBS** | 61.59 (8.79) A | 70.89 (15.13) B | 82.41 (16.65) C | 20.61 | < .001 | .245 |
| **K-r** | 44.16 (7.92) A | 40.80 (6.59) A, B | 38.62 (9.40) B | 3.08 | .049 | .046 |

*Note*. For each line, different letters indicate a significant difference between columns.

problem of generalization outside the sample used to develop the model. Clearly, the avoidance of cross-validation yields results that are overoptimistic and that may not replicate when the model is applied to out-of-sample data. This result was recently confirmed by Bokhari and Hubert [52] when they re-analysed the results of the MacArthur Violence Risk Assessment Study using ML tree models and cross-validation. Also, Pace et al. (2019) [53], in discussing the results of the b test [54] (a test for detecting malingered cognitive symptoms), similarly observed that a decision rule developed on the whole dataset yielded a classification accuracy of 88% on the whole dataset; however, after 10-fold cross-validation, the accuracy dropped to 66%. For the reasons reported above, in the present study, all ML analyses were conducted using 10-fold cross-validation methods that previous research had shown to be robust in replication studies.

The identification of the most informative attributes (or features, or predictors), called "feature selection," is a widely used procedure in ML [55]. The feature selection is a very powerful means to build a classification model that can detect accentuators and symptom producers as accurately as possible. In fact, it permits to remove redundant and irrelevant features, increasing the model generalization and reducing overfitting and noise in the data. In order to identify the most discriminating features for classification, we ran a trial and error procedure using random forest as model. This model consists of many decision trees, each built from a random extraction of observations from the dataset and a random extraction of features. The random extraction is repeated many times, finally selecting the set of features that maximizes the model accuracy. The selected features at the top of trees are generally more important than those selected at end nodes, because the top splits typically produce larger information gains. Following this, we list the most important features for classification accuracy. Based on the analysis, the predictors used to develop the ML models were age, neurological impairment, affective disorders, psychosis, SIMS (TS), low intelligence (LI), amnestic disorders (AM), F-r, Fp-r, Fs, and RBS.

These 11 features were entered into different ML classifiers, which were trained (using a 10-fold cross-validation procedure) to classify every subject as belonging to one of the three groups of interest (Consistent Participants, Symptom Accentuators, Symptom Producers). To ensure that results are stable across different classifiers, not depending on the specific model assumptions, we selected the following classifiers as representative of different categories (from regression to classification trees, to Bayesian statistics): naïve bayes, logistic regression, simple logistic, support vector machine, and random forest (WEKA Manual for Version 3.7.8) [56]. The results and accuracies among different classifiers, as measured by the percentage of participants correctly classified, AUC, and F1 score, are reported in Table 4.

AUC stands for area under the curve in ROC analysis and the F1 score is defined as the weighted harmonic mean of the precision and recall of the test (note that precision) is the

**Table 4. Accuracies of the five ML classifiers as measured by percentage of participants correctly classified, AUC, and F1.**

| Classifier | Accuracy (%) | AUC | F1 |
|---|---|---|---|
| Naïve Bayes | 71.79% | 0.85 | 0.71 |
| Logistic Regression | 70.94% | 0.84 | 0.71 |
| Simple Logistics | 66.67% | 0.83 | 0.66 |
| Support Vector Machine | 69.23% | 0.81 | 0.69 |
| Random Forest | 71.79% | 0.86 | 0.72 |

*Note*. Perfect classification would be equivalent to AUC = 1 and the F1 score = 1.

number of correct positive results divided by the number of all positive results returned by the classifier, and recall (r) is the number of correct positive results divided by the number of all relevant samples, or all samples that should have been identified as positive).

All classifiers were based on different assumptions and representative of different classes of classifiers. However, they all yielded similarly accurate results (in the range of 66.67–71.79%). ML models, such as those reported above, are difficult to interpret: the operations computed by the algorithm to identify the single participant as Consistent, Symptoms Accentuator, or Symptoms Producer are unclear. To better understand the logic on which the classifications results are based, a simpler model, called OneR, was run [57]. This classifier is clearer in terms of transparency of the operations computed by the algorithm and it permits to highlight easily the classification logic. The accuracy of this model was 66.67%, and it followed the following rules:

- *if the SIMS score is < 13.5, then the subject is a consistent participant;*

- *if the SIMS score is < 18.5 or ≥ 34.5, then the subject is an accentuator; and*

- *if the SIMS score is < 34.5, then the subject is a symptom producer.*

According to the classification process followed by the OneR algorithm, amongst the parameters considered, the SIMS score emerged the feature on which the algorithm based its classification efficiency. According to the aforementioned classification rules, indeed, OneR identified cut-off for the SIMS score (i.e., 13.5, 18.5, and 34.5) to distinguish symptom producers from symptom accentuators and consistent participants.

## Discussion

In forensic damage evaluations, it is not always easy to determine whether a given symptom presentation is bona fide or non-credible. For this reason, researchers have designed tests to detect feigning and included, in personality and psychopathological inventories, validity scales to identify what Paulhus defined as "responding bias" [58]—the systematic tendency to answer self-report items in a way that interferes with accurate self-presentation. However, such measures are unable to distinguish between persons who exaggerate existing symptoms and persons who completely fabricate symptoms. In the context of damage evaluations, the ability to differentiate between consistent participants, accentuators, and symptom producers can assist courts in determining the appropriate rates of damage and proportional indemnity. Thus, the present study sought to identify criteria for the identification of accentuators. We analyzed differences in the SIMS and MMPI-2-RF validity scale scores among participants previously classified as consistent participants, accentuators and symptom producers. The SIMS is a widely used tool for identifying feigning, while the MMPI-2-RF validity scales are used to investigate responding bias.

The results for the SIMS indicated that the TS, NI, and LI scales were able to distinguish among consistent participants, accentuators and symptom producers. In contrast, the AF scale could discriminate between consistent participants and symptom producers but was unable to identify whether feigners were exaggerating or fabricating their symptoms. The P and AM scales were able to distinguish between symptom producers and consistent participants/accentuators. Finally, the TS provided an overall estimate of the likelihood that a respondent was fabricating or exaggerating symptoms of psychiatric or cognitive dysfunction. These results not only confirm the findings reported in the literature [26,27,31,33] that TS is one of the best overall indicators of feigning, but they also suggest that TS is capable of distinguishing between accentuators and symptom producers.

In this research, we used the traditional cut-off value of $\geq 14$, which has been proven to show remarkably high sensitivity [26,27] and good specificity. We observed that accentuators obtained scores just over the cut-off value ($M$ = 16.50), whereas symptom producers obtained significantly higher scores ($M$ = 23.15). We argue that the presentation of different optimal cut-off values in the literature ($\geq 14$ [26]; $\geq 19$ [33], and even $\geq 24$ [41]) might reflect a difference in these two subtypes of inconsistent behavior that speak to possible malingering. Considering that a lower cut-off value (e.g., $\geq 14$ or $\geq 16$) might incorrectly identify an accentuator as a symptom producers, whereas a higher cut-off value (e.g., $\geq 19$ or $\geq 24$) might incorrectly identify an accentuator as an consistent respondent, it would be better to use two different cut-offs to identify accentuators and symptoms , respectively.

The NI scale reflects the degree to which a respondent endorses illogical or highly atypical neurological symptoms. Our results showed that the NI scale was not only useful in discriminating between feigners and consistent respondents, but also in differentiating between the two subtypes of inconsistent behavior that speak to possible malingering: accentuators' scores ($M$ = 2.61) were quite close to the cut-off value recommended by Smith and Burger ($\geq 2$), whereas symptom producers obtained significantly higher scores ($M$ = 3.92). The LI scale reflects the degree to which a respondent endorses cognitive incapacity or intellectual deficits inconsistent with the capacities and knowledge typically present in individuals with cognitive or intellectual deficits. In the present study, the LI scale distinguished between consistent respondents ($M$ = 1.06), accentuators ($M$ = 2.11), and symptom producers ($M$ = 4.31) when the recommended cut-off value ($\geq 2$) was used. Again, the LI scale was not only able to identify feigners, but it was also able to distinguish between accentuators (whose scores were distributed around the cut-off value) and symptom producers (who obtained significantly higher scores). In summary, it would be useful to set two cut-off values for the aforementioned SIMS scales for use in identifying accentuators.

The P scale reflects the degree to which a respondent endorses unusual psychotic symptoms that are not typically present in actual psychiatric patients. In this study, using the recommended cut-off value ($\geq 1$), we found that the P scale could not distinguish between consistent respondents ($M$ = 0.90) and accentuators ($M$ = 1.55); however, it could distinguish both of these two groups from symptom producers ($M$ = 2.67). It is worth mentioning that, even though the P scale could identify up to 91.5% of participants who malingered psychotic symptoms [27], individuals with an alleged adjustment disorder associated with depression and anxiety obtained scores past the cut-off value, indicating endorsement of psychotic symptoms. This result is in line with the literature [40], which indicates that malingerers tend to overgeneralize their symptoms.

The AM scale reflects the degree to which a respondent endorses symptoms of memory impairment that are inconsistent with patterns of impairment seen in brain dysfunction or injury. The scale demonstrates high sensitivity for its target psychopathology [33,59,60]. In the present study, we used the recommended cut-off value ($\geq 2$) and found that the AM scale behaved like the P scale: it could not distinguish between consistent respondents ($M$ = 1.35) and accentuators ($M$ = 2.16); however, it could separate both of these groups from symptom producers ($M$ = 3.82).

The AF scale reflects the degree to which a respondent endorses atypical feelings and symptoms of depression and anxiety. This scale also shows high sensitivity for its target psychopathology [33,59,60], correctly identifying up to 100% of individuals who malinger depression when the recommended cut-off value ($\geq 5$) is used [27]. In our study, using a cut-off value of $\geq 5$, AF successfully distinguished between consistent participants ($M$ = 5.73) and feigners, but it was unable to discriminate between accentuators ($M$ = 8.14) and symptom producers ($M$ = 8.46). Despite its high sensitivity, the scale has been criticized because it overlaps with

genuine depressive symptoms [61], thus increasing the risk of false positives [27], as observed in our study.

Our results on the MMPI-2-RF validity scales indicated that the F-r and RBS scales successfully discriminated between consistent individuals, accentuators, and symptom producers. The F-r scale is comprised of 32 items designed to detect unusual or infrequent responses in the normative population. High scores indicate the overreporting of a large range of psychological, cognitive, and somatic symptoms. According to Sellbom and Bagby [62], F-r was designed to more broadly identify infrequent responses across populations; thus, it is likely to work better when less severe psychopathology is overreported, as was the case in the present study, in which participants tended to amplify anxious and depressive symptoms. The RBS scale is comprised of 28 items that measure overreporting as an unusual mix of responses associated with non-credible memory complaints [34]. For both scales, we observed lower scores in consistent participants relative to symptom producers, with the largest effect size in discriminating between accentuators and symptom producers. These results are consistent with the findings of Wygant et al. [63,64], which underlined that F-r and RBS perform best in predicting malingering criteria.

The Fs and Fp-r scales were able to distinguish between symptom producers and consistent respondents/accentuators. The Fp-r scale's 21 items analyze infrequent responses within psychiatric inpatient samples. An elevated score indicates an individual's self-unfavorable reporting and exaggerated psychopathology. In particular, the scale focuses on identifying symptoms that are rarely reported among bona-fide patients with mental illness [65]. The Fs scale is comprised of 16 items describing somatic symptoms that are infrequently observed in medical patient populations. A high score suggests feigning. Both the Fp-r and the Fs scale differentiated symptom producers from the other two groups and showed that, contrary to symptom producers, accentuators did not feign psychotic psychopathology; also, contrary to consistent participants, accentuators did not inflate somatic symptoms more frequently.

The FBS-r scale could distinguish between consistent respondents and feigners, regardless of whether they exaggerated or fabricated their symptoms. The scale was designed for application in a forensic, rather than a clinical, context, and it is comprised of 31 items that define somatic and cognitive symptoms that are rarely reported by personal injury claimants. A high score is associated with overreporting. Specifically, Fs and FBS-r focus on detecting non-credible somatic and/or neurocognitive complaints [66]. Our classification results were consistent with the interpretative guidelines recommended by Ben-Porath and Tellegen [67,68] with regard to suspected symptom exaggeration at T-scores of 90 for F-r, 70 for Fp-r, and 80 for Fs and FBS-r. Similar conclusions were also found by Wygant et al. [69] and Gervais et al. [70] for RBS cut-off values of 80.

Finally, the K-r scale could discriminate between consistent participants and symptom producers but could not differentiate either of these groups from accentuators.

## Conclusion

This preliminary study yielded encouraging results, highlighting that some scales of the SIMS (TS, NI, and LI) and MMPI-2-RF (F-r and RBS) were able to discriminate between consistent participants, accentuators, and symptom producers. The idea behind this research was to identify objective indicators not only to discriminate between consistent and inconsistent test-takers, but also to distinguish between different degrees of inconsistency (exaggeration vs. fabrication), with the aim to offer practitioners and clinicians an empirical-based tool to perform their assessment.

One of the main limitations of the study is the sample size: a small sample increases the likelihood of a type II error skewing the results. Another limitation is the fact that mental health

professionals' evaluations used in this study are characterized by a certain degree of subjectivity; therefore, they could partially be a product of biases and beliefs of the professionals who took part in the research. A further limit of this study concerns the SIMS and the rationale behind its creation: this tool was developed for forensic screening, and it thereby covers a number of feigned dysfunctions that are commonly encountered in criminal proceedings (e.g., intellectual disability, psychotic disorder, amnestic syndromes), wherein defendants might aim at obtaining a diminished capacity plea. Accordingly, the SIMS concerns extreme dysfunctions with lower base rates outside of criminal settings (e.g., a damage evaluation setting), wherein milder and more moderate impairments that are not specifically addressed by the SIMS are more common [40].

Future research should use a computerized version of these tests, enabling researchers to also record behavioral indicators (e.g., reaction time, mouse trajectories); such data has been demonstrated to be useful in faking-good research [38,71–79]. Moreover, future research could implement experimental designs using other tests and questionnaires, exploring empirical differences in scenarios concerning not only feigned mental disorders but also feigned cognitive impairment.

## Author Contributions

**Conceptualization:** Cristina Mazza, Franco Burla, Stefano Ferracuti, Paolo Roma.

**Data curation:** Marco Colasanti.

**Formal analysis:** Cristina Mazza, Graziella Orrù, Merylin Monaro, Marco Colasanti.

**Methodology:** Cristina Mazza, Stefano Ferracuti, Paolo Roma.

**Supervision:** Franco Burla, Merylin Monaro, Stefano Ferracuti, Paolo Roma.

**Writing – original draft:** Cristina Mazza, Graziella Orrù, Merylin Monaro, Marco Colasanti.

**Writing – review & editing:** Franco Burla, Paolo Roma.

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
