## [Decision Letter · Decision Letter 0]

1 Oct 2019

PONE-D-19-22331

SEARCHING FOR INDICATORS TO DISTINGUISH ACCENTUATORS FROM SIMULATORS: A PRELIMINARY STUDY ON MALINGERING SUBTYPES USING THE SIMS AND MMPI-2-RF

PLOS ONE

Dear Dr Roma,

Thank you for submitting your manuscript to PLOS ONE. After careful consideration, we feel that it has merit but does not fully meet PLOS ONE’s publication criteria as it currently stands. Therefore, we invite you to submit a revised version of the manuscript that addresses the points raised during the review process.

We would appreciate receiving your revised manuscript by Nov 15 2019 11:59PM. To enhance the reproducibility of your results, we recommend that if applicable you deposit your laboratory protocols in protocols.io, where a protocol can be assigned its own identifier (DOI) such that it can be cited independently in the future. For instructions see: http://journals.plos.org/plosone/s/submission-guidelines#loc-laboratory-protocols

We look forward to receiving your revised manuscript.

Kind regards,

Stephan Doering, M.D.

Academic Editor

PLOS ONE

1. We note that you have indicated that data from this study are available upon request. PLOS only allows data to be available upon request if there are legal or ethical restrictions on sharing data publicly. For more information on unacceptable data access restrictions, please see http://journals.plos.org/plosone/s/data-availability#loc-unacceptable-data-access-restrictions.

Reviewers' comments:

Reviewer's Responses to Questions

**Comments to the Author**

1. Is the manuscript technically sound, and do the data support the conclusions?

Reviewer #1: Partly

Reviewer #2: Partly

2. Has the statistical analysis been performed appropriately and rigorously? 

Reviewer #1: Yes

Reviewer #2: No

3. Have the authors made all data underlying the findings in their manuscript fully available?

Reviewer #1: Yes

Reviewer #2: Yes

4. Is the manuscript presented in an intelligible fashion and written in standard English?

Reviewer #1: Yes

Reviewer #2: Yes

5. Review Comments to the Author

Reviewer #1: Malingering detection is an important issue in clinical and forensic settings that should be based, not on clinical impressions or judgements, but on the exigencies of a reliable technique grounded on replicable empirical findings. In the evaluation, a multimethod approach is required combining clinical interviews and psychometric instruments, of which the MMPI is the most extensively used (Graham, 2011, Greene, 2011, McDermott, 2012, Osuna et al. 2015, Rogers et al., 2003, Vilariño et al. 2013). So, the aim of this study is very interesting. As the authors comment, it is a preliminary study. However, before it can be published it would be desirable to introduce important changes to improve this paper.

It would be desirable to modify the title since it is excessively generic and the study has been conducted in individuals diagnosed of adjustment disorder with mixed anxiety and depressed mood.

The introduction requires a better organization to keep a sequence in explaining malingering. For example, in my opinion paragraph in page 5, line 93 should be on page 4 line 79. Also, a better explanation on instruments to detect malingering should be included with the correspondent references.

Also, a better structure is required for the materials and method section. There are several questions no clarified in the paper. Information on the participants, sample size, aetiology of psychological damage, and reasons why they are assessed in the Laboratory of Clinical Psychology must be included. Logically the distribution in each of the three established groups is subject to bias. Table 1 can be deleted as it does not provide more information than is included in the text.

The conclusions section should be improved and not repeating the results obtained.

Minor revisions:

In the results section, at all times the decimals must be represented with points (Tables 3 and 4).

Page 17, line 371: To delete the initial ‘.

Reviewer #2: The manuscript entitled, " Searching for indicators to distinguish accentuators from simulators: a preliminary study using the SIMS and the MMPI 2 RF," uses an interesting methodology and advanced statistics to discern possible test markers of partial and full malingering. There are some difficulties with the terminology used that needs to be addressed. The English is general well written, but I do make some suggestions below. I have major concerns about the methods section as written, the results, and the conclusions, but all should be correctable.

Title. Difficulties that I have with the title illustrate some of the major points of correction for the paper. Remove "Searching for." calling the symptom exaggeration group accentuators is fine. Simulators makes sense as presented, but the research design in this field uses simulators compared to controls, with simulators given instructions to feign exaggeration for monetary gain. Therefore, it might be best to find another term, such as symptom producers. If so, perhaps the accentuators could be called symptom accentuators. Better yet, try partial symptom accentuators compared to full symptom producers. I would not use partial and full malingerers as the terms for the following reason. It is not clear to me that the study is actually getting at malingering. There is no valid basis for attributing malingering in full or in part based on the types of inconsistencies used to differentiate the groups. Of course, the goal is to have the study lead to better indicators of full and partial malingerers, and that could go as the long term objectives of the research program along the indicated lines, but this first study is aimed at getting initial test markers of different degrees of inconsistencies as defined operationally by the study. Granted, there are three groups of patients in the study. The first is called presumably honest. I suggest referred to this group as Fully Consistent or Consistent, and the other two groups as Moderately Inconsistent as Excessively Inconsistent. Continuing with the title, use pilot study, not preliminary study. Use Inconsistency subtypes rather than malingering subtypes.

Abstract. Aside from changes occasioned by the above comments, use "two types of inconsistent behavior that speak to possible malingering." Use "differentiate the groups of [your 3 new labels]. Use undergone instead of requested. Use psychiatric/ psychological damage instead of psychic damage. Psychic has a particular meaning in English that does not apply here. Use "scales discriminated among [your 3 labels].

Introduction. Aside from whatever else applies related to my comments about the title and abstract, for line 55, use "Forensic evaluators are trained to evaluate whether evaluees." For that paragraph, the DSM-5 definition of malingering is not the standard of practice one because of its mention of antisocial PD and medico-legal context. You can safely remove lines 59-62 on that. For line 69, I would also check Rogers 2018 book, mentioned in the discussion. Here is a major correction. For the paragraph beginning on line 70, the literature presented is biased toward finding an elevated base rate of malingering in the forensic disability and related context. The Rogers 2018 book referred to the review by Young on the matter in a complementary way. That article was written in 2014 in Psychological Injury and Law, referring to the base rate in the field as 15 +/- 15 % instead of the exaggerated 40 +/- 10%. If you do not want to cite that article, at least cite the references therein that led to the conclusions offered. Start a new paragraph at line 79, and it should begin with "Researchers have." On line 87, refer to the SIRS-2, as well as the SIRS. Line 93. "classifications of and tests related to malingering." Line 110. There are no hypotheses. This seems a serious oversight. Acknowledge such and the reasons why. Presumably the tests were chosen because they were the best available for the task, and there were inklings about which scales would work best. The SIMS total scale makes sense in this regard. As far as I know, the most sensitive scale for the question at hand for the MMPI 2 RF should be the Fp-r, unlike what is suggested later in the paper. There should be a paragraph on the sensitivity of these different scale scores and which ones were expected to be most discriminative. If there were no inklings this way, then indicate such. Were any stats used that have to consider 2-tailed vs. one-tailed testing?

Methods. You refer to Participants but also subjects later on. Check the APA publication manual on this. PS The new one is coming out in October. Line 116. Certified. Use clinical. Line 122 and throughout. Explain exactly what you mean by blind, and in a separate sentence. Line 123. Confirming. Use establishing. First. Second. Third. Add (a) (b) (c). Line 124. Expert on. Use mental health professional with the required training to diagnose." In the English legal world, expert refers to a court designated expert on a matter. Congruences a, b, c. I am requesting a major addition. These congruences need to be explained much better. Were there pre-established criteria; give examples, how often were there disagreements that had to be resolved, etc. Line 131. Conclusion. Use determination. Line 133. conclusions. Line 193. T not t. Here and throughout. Line 194. What exactly was the p after the Bonferroni adjustment?

Results. Line 204. Should no the p not stop at 0? And throughout. Table 3. SD not DS. Astericks a,b, c are not defined. Sometimes the European comma used sometimes the American period. Use the period. The use of the Total SIMS score in the same analysis as the five subscales introduces collinearity. Perhaps run checking stats or run another analysis, adjusting p accordingly. Table 4, aside from what has been mentioned for Table 3, L-r is included here, but not mentioned in the text to this point. Moreover, as mentioned in the discussion, it is not part of an expected parameter for the question at hand. Remove and redo the analysis. Line 238. Page number for the quote. Line 252. Reference for the b test, please. Line 259. Awkward title. Line 262. KPI not defined. Line 265. because, not as. Line 265. Larger, not bigger. Line 267-268. Explain from which tests the predictors come from; for example, psychosis; also is it not amnestic memory and not memory disorder. Line 269. entrained instead of trained, not sure about that though. Line 274. FI not defined. Line 283. I am not an expert on MI and I suspect the same for many readers. Please explain the difference how predictors and classification efficiency variables are chosen in MI. How is it possible that the SIMS variables were best for the latter? How much better were they than the MMPI 2 RF ones, given that this test is the better one to use in forensic disability assessments? Generally for all statistical testing, were distributions of variables tested for normality, homoskedacity, etc., were tests robust for these considerations, etc?

Discussion. Line 302. The Paulus definition should go at the beginning. It illustrates that all that may be investigated is degree of response bias than anything like malingering, per se. The authors should check Rogers and Bender (2018) for all relevant terms and indicate which applies best. Also, check their design suggestions and comment in a future research section, which was not attempted. Line 304. Among, not between. Line 306. Here, the attribution of malingering is given directly. The evidence does not support using this term for those groups, as indicated right from the start of my review. Line 313. The 14 cut-off value for the SIMS in the manual should no longer be used because of its poorer psychometric properties compared to 17. I believe 2 studies in 2014 showed that. You had referred to other possible cut-offs too in your literature review of this test. Moreover, you do not mention cut-offs in any way on this test in the design or results section, so I am confused. You might have to rerun analyses. Line 315. Same problem. Perhaps under instead of over. Line 320. Better to use two. Line 324. but also. And throughout. Differentiated. Use in differentiating. Line 325. There's the malingering word again. Line 334. accentuators from simulators, or whatever new labels you use. Line 338. distinguish, not separate. Line 385. Fs, not FS. Line 388. Inappropriate cut-scores. Check the manual carefully. These levels are suspect only and the second or third levels, not the first. I know you want to help catch people who take advantage of the system, like we all do. But it should not be at the expense of potentially honest respondents by suggesting modification of more stringent criteria, even indirectly. Please revise accordingly. Elaborate more on the limitations of the study. How can future research carry on with the question at hand. Overall, many suggestions, but all doable. Thanks for submitting.

6. PLOS authors have the option to publish the peer review history of their article (what does this mean?). If published, this will include your full peer review and any attached files.

Reviewer #1: No

Reviewer #2: Yes: GERALD YOUNG

---

## [Author Response · Author response to Decision Letter 0]

23 Nov 2019

Reviewer 1

Malingering detection is an important issue in clinical and forensic settings that should be based, not on clinical impressions or judgements, but on the exigencies of a reliable technique grounded on replicable empirical findings. In the evaluation, a multimethod approach is required combining clinical interviews and psychometric instruments, of which the MMPI is the most extensively used (Graham, 2011, Greene, 2011, McDermott, 2012, Osuna et al. 2015, Rogers et al., 2003, Vilariño et al. 2013). So, the aim of this study is very interesting. As the authors comment, it is a preliminary study. However, before it can be published it would be desirable to introduce important changes to improve this paper. 

Thank you for reviewing our manuscript. In the following points, you will find our answers to your suggestions. We have also highlighted our revisions in the text in yellow. 

1. It would be desirable to modify the title since it is excessively generic and the study has been conducted in individuals diagnosed of adjustment disorder with mixed anxiety and depressed mood.

1. According with your suggestion and advices of Reviewer2, we have changed the title as follow: “Indicators to distinguish symptoms accentuators from symptoms producers in individuals with a diagnosed adjustment disorder: a pilot study on inconsistency subtypes using SIMS and MMPI-2-RF”.

2. The introduction requires a better organization to keep a sequence in explaining malingering. For example, in my opinion paragraph in page 5, line 93 should be on page 4 line 79. Also, a better explanation on instruments to detect malingering should be included with the correspondent references.

2. We thank the reviewer for this comment. We agree that the suggested organization is clearer in offering an explanation of malingering and we have re-organized the introduction accordingly. We also added an explanation of instruments to detect malingering, as requested.

3. Also, a better structure is required for the materials and method section. There are several questions no clarified in the paper. Information on the participants, sample size, etiology of psychological damage, and reasons why they are assessed in the Laboratory of Clinical Psychology must be included. Logically the distribution in each of the three established groups is subject to bias. 

3. We added all the required information, especially about the Participants, in Material and Method section.

4. Table 1 can be deleted as it does not provide more information than is included in the text.

4. We have deleted Table 1, as suggested.

5. The conclusions section should be improved and not repeating the results obtained.

5. We agree with the reviewer and we improved the conclusions section including research’s limits and future directions.

Minor revisions

1. In the results section, at all times the decimals must be represented with points (Tables 3 and 4).

1. We apologize for the mistake. We’ve replaced the commas with points in Tables 3 and 4.

2. Page 17, line 371: To delete the initial ‘.

2. We’ve deleted the initial ‘ at page 17 (line 371), thank you.

 

Reviewer 2

The manuscript entitled, "Searching for indicators to distinguish accentuators from simulators: a preliminary study using the SIMS and the MMPI 2 RF," uses an interesting methodology and advanced statistics to discern possible test markers of partial and full malingering. There are some difficulties with the terminology used that needs to be addressed. The English is general well written, but I do make some suggestions below. I have major concerns about the methods section as written, the results, and the conclusions, but all should be correctable.

Thank you very much for reviewing our manuscript and to appreciate out preliminary work. We’ve revised the manuscript trying to address your suggestions and concerns. In the following points, you will find our answers. We have also highlighted our revisions in the text in yellow.

1. Title 

Difficulties that I have with the title illustrate some of the major points of correction for the paper. Remove "Searching for" calling the symptom exaggeration group accentuators is fine. Simulators makes sense as presented, but the research design in this field uses simulators compared to controls, with simulators given instructions to feign exaggeration for monetary gain. Therefore, it might be best to find another term, such as symptom producers. If so, perhaps the accentuators could be called symptom accentuators. Better yet, try partial symptom accentuators compared to full symptom producers. I would not use partial and full malingerers as the terms for the following reason. It is not clear to me that the study is actually getting at malingering. There is no valid basis for attributing malingering in full or in part based on the types of inconsistencies used to differentiate the groups. Of course, the goal is to have the study lead to better indicators of full and partial malingerers, and that could go as the long term objectives of the research program along the indicated lines, but this first study is aimed at getting initial test markers of different degrees of inconsistencies as defined operationally by the study. Granted, there are three groups of patients in the study. The first is called presumably honest. I suggest referred to this group as Fully Consistent or Consistent, and the other two groups as Moderately Inconsistent as Excessively Inconsistent. Continuing with the title, use pilot study, not preliminary study. Use Inconsistency subtypes rather than malingering subtypes.

1. We thank the reviewer for this precious suggestion. We changed the title accordingly: “Indicators to distinguish symptoms accentuators from symptoms producers in individuals diagnosed of adjustment disorder: a pilot study on inconsistency subtypes using SIMS and MMPI-2-RF”. Following your advice, we also adapt the running head as “Distinguish symptoms accentuators from symptoms producers”.

2. Abstract

Aside from changes occasioned by the above comments, use "two types of inconsistent behavior that speak to possible malingering." Use "differentiate the groups of [your 3 new labels]. Use undergone instead of requested. Use psychiatric/ psychological damage instead of psychic damage. Psychic has a particular meaning in English that does not apply here. Use "scales discriminated among [your 3 labels].

2. We adjusted the abstract following your suggestions. We used “symptoms accentuators, symptoms producers and consistent participants” as labels, in order to facilitate readers identification of research groups.

3. Introduction

3.1 Aside from whatever else applies related to my comments about the title and abstract, for line 55, use "Forensic evaluators are trained to evaluate whether evaluees." 

3.2 For that paragraph, the DSM-5 definition of malingering is not the standard of practice one because of its mention of antisocial PD and medico-legal context. You can safely remove lines 59-62 on that. 

3.1-3.2 We adjusted the Introduction section accordingly.

3.3 For line 69, I would also check Rogers 2018 book, mentioned in the discussion. Here is a major correction. 

3.4 For the paragraph beginning on line 70, the literature presented is biased toward finding an elevated base rate of malingering in the forensic disability and related context. The Rogers 2018 book referred to the review by Young on the matter in a complementary way. That article was written in 2014 in Psychological Injury and Law, referring to the base rate in the field as 15 +/- 15 % instead of the exaggerated 40 +/- 10%. If you do not want to cite that article, at least cite the references therein that led to the conclusions offered. 

3.3-3.4 We thank the Reviewer to point out this suggestion. We check Rogers book (2018) both for epidemiology and labels.

3.5 Start a new paragraph at line 79, and it should begin with "Researchers have." 

3.6 On line 87, refer to the SIRS-2, as well as the SIRS. 

3.7 Line 93. "classifications of and tests related to malingering." 

3.5-3.7 We have re-organized the introduction section. We referred to the SIRS-2, as well as the SIRS.

3.7 Line 110. There are no hypotheses. This seems a serious oversight. Acknowledge such and the reasons why. 

3.7 Reviewer is obviously right. We didn’t specify the hypotheses in order to give a more explorative framework to our research. However, following your suggestion, we detailed H1 and H2 at the end of Introduction section.

3.8 Presumably the tests were chosen because they were the best available for the task, and there were inklings about which scales would work best. The SIMS total scale makes sense in this regard. As far as I know, the most sensitive scale for the question at hand for the MMPI 2 RF should be the Fp-r, unlike what is suggested later in the paper. There should be a paragraph on the sensitivity of these different scale scores and which ones were expected to be most discriminative. If there were no inklings this way, then indicate such. Were any stats used that have to consider 2-tailed vs. one-tailed testing?

3.8 We thank the reviewer for this suggestion. We added paragraphs with these data for both instruments. 

4. Methods 

4.1 You refer to Participants but also subjects later on. Check the APA publication manual on this. PS The new one is coming out in October. 

4.1 We thank the reviewer and check the differences between “subjects” and “participants” on APA website (https://apastyle.apa.org/learn/faqs/subjects-and-participants) and changed the label accordingly.

4.2 Line 116. Certified. Use clinical. 

4.3 Line 122 and throughout. Explain exactly what you mean by blind, and in a separate sentence. 

4.4 Line 123. Confirming. Use establishing. First. Second. Third. Add (a) (b) (c). 

4.5 Line 124. Expert on. Use mental health professional with the required training to diagnose." In the English legal world, expert refers to a court designated expert on a matter. 

4.2-4.5 We explained what we mean by “blind”: “Until the end of this evaluation step, the two mental health professionals didn’t have any knowledge regarding the assessment made by the other colleague on the same participant (i.e. blind procedure)”. We also modified the Methods section accordingly the other suggestions.

4.6 Congruences a, b, c. I am requesting a major addition. These congruences need to be explained much better. Were there pre-established criteria; give examples, how often were there disagreements that had to be resolved, etc. 

4.6 We agree with the reviewer’s observation and deepened with examples the congruence evaluation process. We also gave more details about the frequency of disagreements between our two experts.

4.7 Line 131. Conclusion. Use determination. 

4.8 Line 133. conclusions. 

4.9 Line 193. T not t. Here and throughout. 

4.7-4.9 We changed the Methods section accordingly.

4.10 Line 194. What exactly was the p after the Bonferroni adjustment?

4.10 We have inserted a column with the p after the Bonferroni adjustment for each F test, both for Table 3 and Table 4 (MANCOVA analyses).

4.11 Results. Line 204. Should no the p not stop at 0? And throughout. 

4.12 Table 3. SD not DS. Astericks a, b, c are not defined. Sometimes the European comma used sometimes the American period. Use the period. 

4.11-4.12 We corrected the mistakes; we apologize for these oversights.

4.13 The use of the Total SIMS score in the same analysis as the five subscales introduces collinearity. Perhaps run checking stats or run another analysis, adjusting p accordingly. 

4.13 We thank the reviewer for raising this concern. We checked the multicollinearity assumption using the correlation matrix, both for one and two tailed (please find attached SPSS outputs). The results were, for every correlation, below the suggested cutoff of .90 (Tabachnick & Fidell, 2012) and even below the more conservative value of .80.

4.14 Table 4, aside from what has been mentioned for Table 3, L-r is included here, but not mentioned in the text to this point. Moreover, as mentioned in the discussion, it is not part of an expected parameter for the question at hand. Remove and redo the analysis. 

4.14 We agree with reviewer’s remark. We redone the analysis without L scale.

4.15 Line 238. Page number for the quote. 

4.16 Line 252. Reference for the b test, please. 

4.17 Line 259. Awkward title. 

4.18 Line 262. KPI not defined. 

4.19 Line 265. because, not as. 

4.20 Line 265. Larger, not bigger. 

4.15-4.20 We made all the requested changes: we inserted the number for the quote at line 238, added reference for b test, deleted the title at line 259 changing the title at line 230, defined KPI and use “because” and “larger” at line 265. 

4.21 Line 267-268. Explain from which tests the predictors come from; for example, psychosis; also is it not amnestic memory and not memory disorder. 

4.21 We inserted a detailed explanation about predictors selection in Machine Learning Models paragraph.

4.22 Line 269. entrained instead of trained, not sure about that though. 

4.22 The term “trained” is correct and it is a typical Machine Learning term.

4.23 Line 274. FI not defined. 

4.23 We have added in the manuscript an extensive definition of F1 (see the note of Table 5). 

4.24 Line 283. I am not an expert on MI and I suspect the same for many readers. Please explain the difference how predictors and classification efficiency variables are chosen in MI. How is it possible that the SIMS variables were best for the latter? How much better were they than the MMPI 2 RF ones, given that this test is the better one to use in forensic disability assessments? Generally for all statistical testing, were distributions of variables tested for normality, homoskedacity, etc., were tests robust for these considerations, etc?

4.24 Many thanks for your comment. Thanks to your observation we have noticed that we mistakenly omitted the SIMS score from the list of the variables selected by the feature selection procedure. It was an oversight, so the total number of features selected was 11, including SIMS score. This mistake generated your confusion between the predictors and classification efficiency variables. Moreover, as we agree to the fact that the ML logic is complex for a not expert reader, we tried to clarify what ML is and how it works, including a deeper explanation of cross-validation and predictors selection. Please, find below the most relevant changes that we made in the manuscript to give more clarity to ML concepts:

“ML can be defined as “the study and construction of algorithms that can learn information from a set of data (called a training set) and make predictions for a new set of data (called a test set). In other words, it consists of training one or more algorithms to predict outcomes without being explicitly programmed and only uses the information learned from the training set.”

“The k-fold cross-validation is a technique used to evaluate predictive models by repeatedly partitioning the original sample into a training set to train the model, and a validation set to evaluate it. Specifically, in this paper we adopted a 10-fold cross-validation procedure, as the original sample was randomly partitioned into 10 equal-size subsamples, the folds. Of the 10 subsamples, a single subsample is retained as validation data for testing the model, and the remaining 10-1=9 sub-samples were used as training data. Such process is repeated 10 times, with each of the 10 folds used exactly once as validation data. The results from the 10 folds were then averaged to produce a single estimation of prediction accuracy.”

“The identification of the most informative attributes (or features, or predictors), called “feature selection”, is a widely used procedure in ML. The feature selection is a very powerful mean to build a classification model that can detect accentuators and simulators as accurately as possible. In fact, it permits to remove redundant and irrelevant features, increasing the model generalization and reducing overfitting and noise in the data. In order to identify the most discriminating features for classification, we run a trial and error procedure using random forest as model. This model consists of many decision trees, each built from a random extraction of observations from the dataset and a random extraction of features. The random extraction was repeated many times, finally selecting the set of features that maximized the model accuracy. The selected features at the top of trees are generally more important than those selected at end nodes, as the top splits typically produce bigger information gains. Following this, we listed the most important features for classification accuracy.”

“ML models, such as those reported above, are difficult to interpret: the operations computed by the algorithm to identify the single participant as accentuator, simulator or honest are unclear. To better understand the logic on which the classifications results are based on, a simpler model called OneR was ran. This classifier is clearer in terms of transparency of the operations computed by the algorithm and it permits to easily highlight the classification logic.”

5. Discussion

5.1 Line 302. The Paulus definition should go at the beginning. It illustrates that all that may be investigated is degree of response bias than anything like malingering, per se. The authors should check Rogers and Bender (2018) for all relevant terms and indicate which applies best. Also, check their design suggestions and comment in a future research section, which was not attempted. 

5.1 We thank the reviewer for sharing the work of Rogers & Bender (2018) “Clinical Assessment of Malingering and Deception 4th edition”. We found many interesting insights and ideas we’d like to implement in future researches, as we have explained in Conclusion section. Moreover, reading the section about “basic concepts and definitions” led us to choose the term feigning instead of malingering, being the correct word for describing the type of response style observed. 

5.2 Line 304. Among, not between. 

5.2 We complied with your request.

5.3 Line 306. Here, the attribution of malingering is given directly. The evidence does not support using this term for those groups, as indicated right from the start of my review. 

5.3 See answer #5.1

5.4 Line 313. The 14 cut-off value for the SIMS in the manual should no longer be used because of its poorer psychometric properties compared to 17. I believe 2 studies in 2014 showed that. You had referred to other possible cut-offs too in your literature review of this test. Moreover, you do not mention cut-offs in any way on this test in the design or results section, so I am confused. You might have to rerun analyses. 

5.5 Line 315. Same problem. Perhaps under instead of over. 

5.4-5.5 Albeit we’re aware of previous research findings in clinical settings [van Impelen et al. (2014); Rogers et al. (2014)], we decided to use the 14 cut-off value because participants’ evaluations were conducted into a medical-legal setting in which we had to refer to the SIMS Italian technical manual [La Marca S., Rigoni, D., Sartori, G., & Lo Priore, C. (2011), Giunti OS, Firenze, p. 17]. We decided to report the other cut-off scores in order to be the readers aware of the existence and the use of SIMS cut-offs higher than 14. 

5.6 Line 320. Better to use two. 

5.7 Line 324. but also. And throughout. Differentiated. Use in differentiating. 

5.8 Line 325. There's the malingering word again. 

5.9 Line 334. accentuators from simulators, or whatever new labels you use. 

5.10 Line 338. distinguish, not separate. 

5.11 Line 385. Fs, not FS. 

5.6-5.11 We followed all these suggestions and changed the terminology as aforementioned clarified. 

5.12 Line 388. Inappropriate cut-scores. Check the manual carefully. These levels are suspect only and the second or third levels, not the first. I know you want to help catch people who take advantage of the system, like we all do. But it should not be at the expense of potentially honest respondents by suggesting modification of more stringent criteria, even indirectly. Please revise accordingly. 

5.12 We checked again these references: a) Ben-Porath, Y.S. (2012), Interpreting the MMPI-2-RF. University of Minnesota Press, pp. 245-250 and b) Ben-Porath, Y.S., & Tellegen, A. (2011), MMPI-2-RF Manuale di istruzioni. Adattamento italiano a cura di Sirigatti, S., & Casale, S. (2012), Giunti OS, Firenze, pp. 49-56 [the Italian Technical Manual].

5.13 Elaborate more on the limitations of the study. How can future research carry on with the question at hand? 

5.13 We thank the reviewer for this suggestion. We improved the conclusions section including research’s limits and future directions.

Overall, many suggestions, but all doable. Thanks for submitting.

Thank you very much for all your inspiring and helpful suggestions.

---

## [Decision Letter · Decision Letter 1]

6 Dec 2019

PONE-D-19-22331R1

INDICATORS TO DISTINGUISH SYMPTOM ACCENTUATORS FROM SYMPTOM PRODUCERS IN INDIVIDUALS WITH A DIAGNOSED ADJUSTMENT DISORDER: A PILOT STUDY ON INCONSISTENCY SUBTYPES USING SIMS AND MMPI-2-RF

PLOS ONE

Dear Dr. Roma,

Thank you for submitting your manuscript to PLOS ONE. After careful consideration, we feel that it has merit but does not fully meet PLOS ONE’s publication criteria as it currently stands. Therefore, we invite you to submit a revised version of the manuscript that addresses the points raised during the review process.

We would appreciate receiving your revised manuscript by January 7, 2020. To enhance the reproducibility of your results, we recommend that if applicable you deposit your laboratory protocols in protocols.io, where a protocol can be assigned its own identifier (DOI) such that it can be cited independently in the future. For instructions see: http://journals.plos.org/plosone/s/submission-guidelines#loc-laboratory-protocols

We look forward to receiving your revised manuscript.

Kind regards,

Stephan Doering, M.D.

Academic Editor

PLOS ONE

Reviewers' comments:

Reviewer's Responses to Questions

**Comments to the Author**

1. If the authors have adequately addressed your comments raised in a previous round of review and you feel that this manuscript is now acceptable for publication, you may indicate that here to bypass the “Comments to the Author” section, enter your conflict of interest statement in the “Confidential to Editor” section, and submit your "Accept" recommendation.

Reviewer #1: (No Response)

Reviewer #2: (No Response)

2. Is the manuscript technically sound, and do the data support the conclusions?

Reviewer #1: Yes

Reviewer #2: Yes

3. Has the statistical analysis been performed appropriately and rigorously? 

Reviewer #1: Yes

Reviewer #2: Yes

4. Have the authors made all data underlying the findings in their manuscript fully available?

Reviewer #1: (No Response)

Reviewer #2: Yes

5. Is the manuscript presented in an intelligible fashion and written in standard English?

Reviewer #1: (No Response)

Reviewer #2: Yes

6. Review Comments to the Author

Reviewer #1: The paper has been improved following the recommendations and I consider it suitable for publication.

Reviewer #2: the authors have responded to most of my points. There are a few left. Line 34; producers, and. 39; damage (or psychological/ psychiatric damage). 96. A study by. 96. scales'. 97; e.g., 98. sensitive. 106. FBS-r. 108. FBS-r. 114 set." 120. have on SIMS. 126. judge's order,. 128. Rome, which. 133-4; 30% were a. 135; episodes, and. 135; originated from domestic. 141; e.g., . 149; e.g., . 153; did not. 154; i.e., . 157; times, while; 157; 23 cases involved mental; 157; reaching. 158; on the third. Table 1; still there; 219; Bonferroni not explained; 237 and 253; letters still not clear; I did figure it out, though; 274; paper, we; 275; procedure, in which the; 277; sub-samples are used; 278; folds are then; 291; selection,". 291; means. 295; ran; 297; extraction is. 298; minimizes; 300; list. 318; the FI319; note that. 326; Accentuator, . 327; based, . 328; model, called OneR, was run. 329; to highlight easily. 335; score emerged the; 336; identified. 337; it seems the results mentioned in the prior sentence are not provided. And would these results change anything in the discussion? 347; accentuators, and. 350; scores among participants. 428; feign. 448; vs. . 452; subjectivity; therefore, they.

7. PLOS authors have the option to publish the peer review history of their article (what does this mean?). If published, this will include your full peer review and any attached files.

Reviewer #1: No

Reviewer #2: Yes: Gerald Young

---

## [Author Response · Author response to Decision Letter 1]

9 Dec 2019

Reviewer 1

The paper has been improved following the recommendations and I consider it suitable for publication.

Thank you for reviewing our manuscript a second time, and to consider it suitable for publication.

Reviewer 2 

The authors have responded to most of my points. There are a few left. 

Thank you for reviewing our manuscript again. In the following points, you will find our answers to your suggestions. We have also highlighted our revisions in the text in yellow. 

Line 34; producers, and. 

39; damage (or psychological/ psychiatric damage). 

96. A study by. 

96. scales'. 

97; e.g., 

98. sensitive. 

106. FBS-r. 

108. FBS-r. 

114 set." 

120. have on SIMS. 

126. judge's order,. 

128. Rome, which. 

133-4; 30% were a. 

135; episodes, and. 

135; originated from domestic. 

141; e.g., . 

149; e.g., . 

153; did not. 

154; i.e., . 

157; times, while; 

157; 23 cases involved mental; 

157; reaching. 

158; on the third. Table 1; still there; 

219; Bonferroni not explained

237 and 253; letters still not clear; I did figure it out, though;

274; paper, we; 

275; procedure, in which the; 

277; sub-samples are used; 

278; folds are then; 

291; selection,".

291; means. 

295; ran; 

297; extraction is. 

298; minimizes; 

300; list. 

318; the FI

319; note that. 

326; Accentuator, . 

327; based, . 

328; model, called OneR, was run. 

329; to highlight easily. 

335; score emerged the; 

336; identified. 

337; it seems the results mentioned in the prior sentence are not provided. And would these results change anything in the discussion? 

347; accentuators, and. 

350; scores among participants. 

428; feign. 

448; vs. . 

452; subjectivity; therefore, they.

We have made all the requested changes, thank you. 

Line 158. We have deleted Table 1 (“Group’s composition according to the criteria set out”) during the first revision, according to Reviewer1’s suggestion. The table that is now entitled Table 1 describes the “Demographic Composition of the Three Research Groups” (line 172).

Line 219. Regarding Bonferroni correction, SPSS automatically compute the p value after the adjustment. “Confidence interval adjustment. Select least significant difference (LSD), Bonferroni, or Sidak adjustment to the confidence intervals and significance” (IBM SPSS Advanced Statistics 26, p.15). Moreover, we have added this sentence in “Statistical Analysis Section”: “The Bonferroni correction was applied to adjust confidence intervals; SPSS-25 software (SPSS Inc., Chicago, IL) automatically corrected the p-value for the number of comparisons.”

Line 337. To greater clarity, we have rephrased the sentence as follow: “According to the aforementioned classification rules, indeed, OneR identified cut-off for the SIMS score (i.e., 13.5, 18.5, and 34.5) to distinguish symptom producers from symptom accentuators and consistent participants.”

---

## [Editor Report · Decision Letter 2]

13 Dec 2019

INDICATORS TO DISTINGUISH SYMPTOM ACCENTUATORS FROM SYMPTOM PRODUCERS IN INDIVIDUALS WITH A DIAGNOSED ADJUSTMENT DISORDER: A PILOT STUDY ON INCONSISTENCY SUBTYPES USING SIMS AND MMPI-2-RF

PONE-D-19-22331R2

Dear Dr. Roma,

We are pleased to inform you that your manuscript has been judged scientifically suitable for publication and will be formally accepted for publication once it complies with all outstanding technical requirements.

With kind regards,

Stephan Doering, M.D.

Academic Editor

PLOS ONE

---

## [Editor Report · Acceptance letter]

18 Dec 2019

PONE-D-19-22331R2 

Indicators to distinguish symptom accentuators from symptom producers in individuals with a diagnosed adjustment disorder: A pilot study on inconsistency subtypes using SIMS and MMPI-2-RF   

Dear Dr. Roma:

I am pleased to inform you that your manuscript has been deemed suitable for publication in PLOS ONE. Congratulations! Your manuscript is now with our production department. 

With kind regards,

on behalf of

Professor Stephan Doering 

Academic Editor

PLOS ONE